# The Relationship between Food Security Status and Sleep Disturbance among Adults: A Cross-Sectional Study in an Indonesian Population

**DOI:** 10.3390/nu12113411

**Published:** 2020-11-06

**Authors:** Emyr Reisha Isaura, Yang-Ching Chen, Hsiu-Yueh Su, Shwu-Huey Yang

**Affiliations:** 1Department of Nutrition, Faculty of Public Health, Airlangga University, Surabaya, East Java 60115, Indonesia; emyr.reisha@fkm.unair.ac.id; 2School of Nutrition and Health Sciences, College of Nutrition, Taipei Medical University, Taipei 11031, Taiwan; melisa26@gmail.com; 3Research Group of Food Safety and Food Security, Faculty of Public Health, Airlangga University, Surabaya, East Java 60115, Indonesia; 4Department of Family Medicine, School of Medicine, College of Medicine, Taipei Medical University, Taipei 11031, Taiwan; 5Department of Family Medicine, Taipei Medical University Hospital, Taipei 11031, Taiwan; 6Department of Dietetics, Taipei Medical University Hospital, Taipei 11031, Taiwan; 7Nutrition Research Center, Taipei Medical University Hospital, Taipei 11031, Taiwan; 8Research Center of Geriatric Nutrition, College of Nutrition, Taipei Medical University, Taipei 11031, Taiwan

**Keywords:** food insecurity, sleep disturbance, adults, cross-sectional study

## Abstract

Background: The relationship between food insecurity and the experience of sleep disturbance has received little attention among researchers, although food insecurity is associated with poor physical and mental health globally. This study aimed to investigate the relationship between food security status and sleep disturbance among adults 20–64 years old. Methods: The study’s population-based sample included 20,212 Indonesian adults who participated in the fifth wave of the Indonesia Family Life Survey (IFLS5) in 2014. Dietary intake data, gathered using a food frequency questionnaire (FFQ), were used to assess the food security status. Sleep disturbance was assessed using the 10-item Patient-Reported Outcomes Measurement Information System (PROMIS) questionnaire. We used multiple linear and logistic regression models to test the study hypothesis. Results: A higher likelihood of experiencing sleep disturbance was recorded in people aged older than 56 years (OR = 1.78, 95% CI: 1.17–2.72, *p* = 0.007), people with depressive symptoms (OR = 3.57, 95% CI: 2.77–4.61, *p* < 0.001), and food-insecure people (OR = 1.32, 95% CI: 1.02–1.70, *p* = 0.036). A lower likelihood of experiencing sleep disturbance was recorded in people with low educational attainment (OR = 0.41, 95% CI: 0.30–0.57, *p* < 0.001). Sleep disturbance was dependent on the food consumption groups and food security status among men (*p* = 0.004). Conclusions: Sleep disturbance may be affected by the food-insecure status of adults, and later, may lead to serious health outcomes.

## 1. Introduction

Food insecurity is a public health problem that exists globally, including in developing countries. Food insecurity is the disruption of dietary patterns or nutrient intake because of a lack of finances and other resources [1]. In the USA, researchers have estimated that food insecurity affects 9–14% of adults aged 24–32 years, and shows higher rates among women and low-income adults [2,3,4]. Meanwhile, developing countries, such as Indonesia, have reported a high prevalence of food insecurity, with as many as 19.4 million people out of a population of 258.7 million who were unable to meet their dietary requirements [5]. Food-insecure people may encounter anxiety or stress from uncertainty about fulfilling their food requirements or other necessities. The anxiety and stress may lead to depressive symptoms and affect their quality of sleep [6,7]. Poor sleep quality, such as short sleep duration, the presence of sleep disturbance, or difficulty getting to sleep, is associated with the possibility of chronic disease later in life [8,9,10]. Sleep disturbance is known to be a big problem in developed countries. In the United States, the National Institutes of Health stated that in 2017, about thirty percent of people reported getting less than seven hours of sleep each night [11]. The problem has begun to slide to low-to-middle-income countries, which reported about ten percent of people having severe sleeping problems [12,13]. People with sleep disturbances have reported difficulties with concentrating during the day and feeling restless, anxious, and depressed [14,15]. On the other hand, depressed people may also experience economic difficulties, which leads to food insecurity [16,17]. People with food insecurity are known to be vulnerable and more likely to develop chronic diseases or experience poorer mental health. However, the relationship between food insecurity and sleep disturbance is rather vague. This study’s objective was to determine the relationship between food security statuses and sleep disturbance using a nationally representative sample of adults aged 20–64 years in Indonesia.

## 2. Materials and Methods

### 2.1. Study Participants and Data Source

This cross-sectional study used the secondary longitudinal data of the fifth wave of the Indonesian Family Life Survey (IFLS5) that was conducted in 2014. The initial survey (IFLS1) was conducted in 1993, representing about 80 percent of the Indonesian population. The IFLS5 datasets were anonymous, included participants of all ages, and were available for researchers who met the criteria based on the RAND Corporation guidelines about the dataset usage [18]. The institutional review board (IRB) review of the IFLS studies went through the sufficient and appropriate review that followed the IRB guidelines and was approved by both the RAND Corporation and Indonesia’s Institutions, in particular, the Survey Meter institution for the IFLS5 study [18,19]. The total number of participants for the IFLS5 were 34,464 people, aged from zero to older than 80 years old. The present study included participants who had complete data relating to food, anthropometric characteristics, sociodemographic characteristics, blood pressure, depressive symptoms, and sleep disturbance. We excluded participants who had been diagnosed with cancer, chronic diseases (e.g., diabetes and cardiovascular diseases), had a disability, or who were breastfeeding or pregnant in order to minimize the probability of sampling bias. Therefore, 20,212 participants aged 20–64 years old were included in the present study.

### 2.2. Assessment of Sleep Disturbance

The assessment of sleep disturbance was based on the guidelines of the Patient-Reported Outcomes Measurement Information System (PROMIS) [20,21]. The IFLS5 used self-reported answers to ten questions based on the PROMIS questionnaire for the assessment of sleep deprivation and sleep quality [18]. Each question on the questionnaire was rated using a one-to-five scale (never, rarely, sometimes, often, and always, respectively). The total score range of the ten questions was then summarized with a range from 10 to 50, which was the so-called total raw score. We used a t-score table to identify the t-score that related to every participant’s total raw score and the information was attached to a t-score row based on its value. The t-scores were interpreted as “none to slight” for the participants with scores <55, “mild” for the participants with scores 55.0–59.9, and a combination of “moderate to severe” for the participants with scores >60 [20,21,22]. For the purposes of this study, we also categorized the data into two groups: participants whose sleep disturbance was “mild or less” or “greater than mild.”

### 2.3. Assessment of Food Security Status

The assessment of food insecurity is associated with a person’s lack of secure access to fulfilling their need for a nutritious diet in a sufficient amount to keep an active healthy life, considering both the food frequency and food diversity [23,24,25]. There are several ways to assess food insecurity at the individual level [26,27,28]. One of the assessments of the food insecurity concept was developed using the food frequency questionnaire (FFQ). The World Food Programme (WFP) introduced this concept to assess food consumption analysis by producing food consumption scores (FCSs) and cut-off points for food insecurity [24,25]. The FCSs allow data to be categorized into three food consumption groups (FCGs) consisting of poor, borderline, and acceptable groups. Furthermore, the ten food items listed on the IFLS5 food frequency questionnaire were included in the food consumption analysis. The IFLS5 FFQ asked about ten food types that were eaten by the participants during the last seven days before the interview. We then grouped these ten food types into five food groups. The first group was the staple group, consisting of sweet potato. The second group was the protein group, consisting of eggs, fish, and meats. The third group was dairy products. The fourth group was the fruit group, consisting of banana, mango, and papaya. The last group was the vegetable group, consisting of green leafy vegetables and carrots [23,24,25]. The score for each food group was then summed as the food consumption score in the form of continuous data. The “poor” food consumption group included participants who had an FCS lower than 21. The “borderline” FCG included participants who had food consumption scores that lay between 21 and 35. The “acceptable” FCG included participants who had food consumption scores higher than 35 [24]. The “poor” FGC and “borderline” FGC participants were defined as food-insecure persons [25,29].

### 2.4. Assessment of Covariates

The covariates of the present study were anthropometric characteristics, blood pressure, physical activity, and sociodemographic measurements. We used the body mass index, and additionally, for the participants aged 40 years and older, the measurements of the waist circumference and the body shape index [30,31,32] were added. Furthermore, the adult body mass indices (in kg/m^2^) adopted the Indonesian cut-off points [33]. The body mass index (BMI) was categorized into “normal weight” for participants with a BMI between 18.5 to 25.0, “overweight” for participants with a BMI between 25.1 to 27.0, and “obese” for participants with a BMI higher than 27.0 [34]. The definition of abdominal obesity used the waist circumference measurement with two cut-off points (for men: >90 cm, for women: >80 cm). Trained nurses performed the anthropometric and blood pressure measurements. For the blood pressure measurements, the participants were in the seated position. Participants were defined as having hypertension if the systolic blood pressure (SBP) was ≥140 mmHg or the diastolic blood pressure (DBP) was ≥90 mmHg, or if they had been diagnosed by paramedics before the interview or were currently consuming blood pressure-lowering medication [32].

Furthermore, physical activity was assessed using the number of days on which respondents had done two types of physical activity (i.e., vigorous and moderate) within the last seven days before the survey. We considered the volume of physical activity (PA) to be a continuous variable in the analysis. Respondents answered the self-reported questionnaires in terms of whether they had engaged in physical activities for at least ten minutes continuously during the last seven days. If the respondents said yes, then they were further asked about the number of days on which they had done each type of physical activity. Eight and four metabolic equivalent of tasks (METs), respectively, were then multiplied by the minutes and days of each type of physical activity to form the physical activity volume (in METs minutes/week) [35]. For example, the vigorous physical activity volume formula was the minutes/day multiplied by days/week of doing vigorous PA multiplied by eight METs.

The sociodemographic variables included smoking habits, educational attainment, living area, and marital status, which were presented as categorical data. The smoking habits of the participants were categorized into never (never had a smoking habit), current smoker (currently has a smoking habit), and former smoker (has stopped a smoking habit). The participants’ educational attainment was categorized into low: <12 years of school attainment, and high: ≥12 years of school attainment.

Depressive symptoms were defined using the score of the questionnaire about mental health. The IFLS5 used the ten-item self-reported questionnaire from the Center for Epidemiologic Studies–Depression (CES-D) to assess the mental health of the adult participants. Some prior researchers have used the 10-CES-D questionnaire to assess adults’ depressive symptoms [36,37]. The form of the responses to the 10-CES-D questionnaire was based on four scale items: less than one day (rarely or never), one to two days (some days), three to four days (occasionally), and five to seven days (most of the time). The scores of the 10 questions were then added, resulting in a score ranging from ten to forty. Furthermore, the score was rebased so that the lowest score was zero and the highest score was thirty. The highest score identified people with the most symptomatology of depression [38]. The cut-off point for defining a person as having a risk of heightened depressive symptoms was a score higher than or equal to ten [39,40].

### 2.5. Statistical Analysis

The present study used secondary data from the IFLS5 (2014). The characteristics of participants were presented as a mean and standard deviation for the continuous data and as numbers with percentages for the categorical data. We used one-way ANOVA for the continuous data, with the Bonferroni post hoc test or chi-squared test for the categorical data to compare the values between groups. Furthermore, we used the regression model to assess the relationship between sleep disturbance and depressive symptoms by food consumption group. Furthermore, we also used the sleep disturbance score and CES-D-10 score as continuous data in the linear regression analysis and as categorical data in the logistic regression analysis. To assess the relations of interest, we used a linear and logistic regression model, which was presented using an exponentiated beta coefficient or odds ratio and a 95% confidence interval, respectively. Moreover, this study used three models that accounted for various potential confounders in the multiple logistic regression model. The three models were an unadjusted model, a model with adjustment for age and sex, and a model with a complete adjustment; the complete adjustment was an adjustment for age, sex, educational attainment, marital status, BMI, living area, blood pressure, smoking habit, and physical activity volume. We used a similar sequence of adjustments for potential confounders, which were also used for the linear regression models. Statistical significance was designated as a *p*-value < 0.05. We conducted a multivariate test to identify the characteristics related to sleep disturbance and depressive symptoms, which were analyzed in a separate model. Covariates in these two models included sex, age group, living area, educational attainment, smoking status, blood pressure, physical activity volume, and body mass index. All the analyses were conducted using STATA statistical software (v16.1; StataCorp LP, College Station, TX, USA).

## 3. Results

This cross-sectional study included 20,212 participants (women = 10,070, men = 10,142) from the IFLS5 dataset (Table 1). The flowchart related to the selection of the participants is shown in Appendix A. The mean age of the participants was 39 (standard deviation (SD): 11) years old. Most of the participants had low educational attainment, were currently or ever married, living in urban areas, and had never had a smoking habit. Additionally, most of the participants aged 40 years and older had abdominal obesity, which was observed in 4494 (51.54%) people. The prevalence of food insecurity in this study was 53.86%, or 10,886 of the total participants. Among the food-insecure participants, women represented as many as 5602 (51.46%) people, low education attainment was found in 6832 (62.76%) people, people who were currently married or had a marriage experience constituted 9707 (89.17%) people, and the number of people who lived in urban areas was 6041 (55.49%). Most of the food-insecure participants, i.e., 6544 (60.11%) people, never had a smoking habit, whereas 3944 (36.23%) people reported that they had quit smoking. We found that 191 (1.75%) food-insecure participants were taking blood-pressure-lowering medication and 61 (0.56%) people were taking cholesterol-lowering medication. Furthermore, 2294 (49.95%) people aged 40 years old and older among the food-insecure groups had abdominal obesity. We found that the number of people with depressive symptoms among the food-insecure group was 3808 (34.98%). Furthermore, 595 (56.67%) of the food-insecure participants were experiencing “mild” sleep disturbance, whereas 168 (62.92%) of the food-insecure participants were experiencing “moderate-to-severe” sleep disturbance. The means of the systolic blood pressures and vigorous physical activity volumes of the food-insecure participants were higher than those in the food-secure group (*p* < 0.001).

Figure 1 shows the prevalence of sleep disturbance by FCG for women and men. Among men, participants reported a “none-to-slight” sleep disturbance level for 94.17% (*n* = 4575) of the acceptable FCG, 93.54% (*n* = 3260) of the borderline FCG, and 92.22% (*n* = 1659) of the poor FCG. Participants reported a “mild” sleep disturbance level for 5.00% (*n* = 243) of the acceptable FCG, 4.99% (*n* = 174) of the borderline FCG, and 6.06% (*n* = 109) of the poor FCG. Furthermore, participants reported a “moderate-to-severe” sleep disturbance level for 0.82% (*n* = 40) of the acceptable FCG, 1.46% (*n* = 51) of the borderline FCG, and 1.72% (*n* = 31) of the poor FCG. The level of sleep disturbance was dependent on the food consumption group or food security status among men (*p* = 0.004).

Among women, participants reported a “none-to-slight” sleep disturbance level for 93.93% (*n* = 4197) of the acceptable FCG, 92.80% (*n* = 3370) of the borderline FCG, and 93.06% (*n* = 1825) of the poor FCG. Participants reported a “mild” sleep disturbance level for 4.74% (*n* = 212) of the acceptable FCG, 5.58% (*n* = 203) of the borderline FCG, and 5.56% (*n* = 109) of the poor FCG. Furthermore, participants reported a “moderate-to-severe” sleep disturbance level for 1.32% (*n* = 59) of the acceptable FCG, 1.62% (*n* = 59) of the borderline FCG, and 1.38% (*n* = 27) of the poor FCG. The level of sleep disturbance was independent of the food consumption group or food security status among women (*p* = 0.301).

Table 2 shows the characteristics related to sleep disturbance, as determined by the regression model. When the confounding variables were taken into account in the multivariate regression model, several characteristics remained independently related to sleep disturbance. A higher likelihood of experiencing sleep disturbance was recorded in participants aged older than 56 years (OR = 1.78, 95% CI: 1.17–2.72, *p* = 0.007), participants with depressive symptoms (OR = 3.57, 95% CI: 2.77–4.61, *p* < 0.001), and food-insecure participants (OR = 1.32, 95% CI: 1.02–1.70, *p* = 0.036). However, a lower likelihood of experiencing sleep disturbance was recorded in participants with low educational attainment (OR = 0.41, 95% CI: 0.30–0.57, *p* < 0.001).

Furthermore, Appendix A shows that a higher likelihood of experiencing sleep disturbance after adjusting for age and gender in our study was reported in participants with depressive symptoms (OR = 3.50, 95% CI: 2.72–4.52, *p* < 0.001) and participants with food insecurity (OR = 1.47, 95% CI: 1.15–1.89, *p* = 0.002). On the other hand, a lower likelihood of experiencing sleep disturbance was reported in participants with low educational attainment (OR = 0.38, 95% CI: 0.28–0.51, *p* < 0.001). Furthermore, Appendix A presents the logistic regression between characteristics related to the sleep disturbance level. A higher likelihood of experiencing sleep disturbance among men in our study was reported in participants with depressive symptoms (OR = 3.16, 95% CI: 2.17–4.60, *p* < 0.001) and participants with food insecurity (OR = 1.75, 95% CI: 1.19–2.58, *p* = 0.005). Meanwhile, a higher likelihood of experiencing sleep disturbance among women in our study was reported in participants with depressive symptoms (OR = 3.16, 95% CI: 2.17–4.60, *p* < 0.001) and in participants aged older than 36 years (OR = 1.73–2.54, 95% CI: 1.00–2.72, *p* = 0.016–0.002). On the other hand, a lower likelihood of experiencing sleep disturbance among men in our study was reported in current smoker participants (OR = 0.35, 95% CI: 0.12–0.99, *p* = 0.048) and in participants with low educational attainment (OR = 0.46, 95% CI: 0.30–0.71, *p* < 0.001). Moreover, a lower likelihood of experiencing sleep disturbance among women in our study was reported in participants with low educational attainment (OR = 0.37, 95% CI: 0.22–0.61, *p* < 0.001).

## 4. Discussion

The present study demonstrates the relationship between food insecurity and sleep disturbance. This study’s results show that most of the food-insecure participants were women, people who had marriage experience, people with low educational attainment, and people who lived in urban areas. In addition, the middle-aged participants (older than 40 years old) were more likely to have abdominal obesity. Furthermore, the means of the systolic blood pressure and vigorous physical activity volume were higher among food-insecure participants compared to food-secure participants. Moreover, participants who were more likely to experience sleep disturbance were aged older than 56 years old, had depressive symptoms, and a food-insecurity status, but this was not the case for the people who had low educational attainment.

Food insecurity is associated with chronic diseases and poor mental health [41,42,43]. The burden of food insecurity and marriage experience, in particular for women, synergistically contributes to the development of depressive symptoms [44]. A potential explanation may be the stressful decision-making that most food-insecure women have to do related to their financial situation, and women may be more susceptible to harm from life stresses and other environmental factors [4,42,45]. The findings from this study also confirmed previous research that suggests the higher prevalence of food insecurity among women [44,46]. Furthermore, our study results are in line with previous findings showing that most people with low education levels are more likely to encounter food insecurity. One of the possible mechanisms is the contribution of the financial hardship of food-insecure people. The financial hardship of food-insecure people may be affected by the low wage or minimum payment from intensive work. Low-educated people are more likely to do manual labor and to receive a minimum payment, which may affect the difficulty in fulfilling their nutrition requirements through adequate meals [47,48]. Another explanation is that low education may lead to less exposure to nutrition education explaining how to maintain a well-balanced and nutritious diet [49].

Food-insecure people who live in the urban areas may try to keep up with their environment, which includes needs relating to food, or cultural or financial situations [50,51,52]. People who live in urban areas might also be exposed to more processed food options [53,54]. The problem with processed foods or fast foods is the lack of consumers’ ability to control the amount of calories, sodium, fat, and sugar [55]. In particular, for people with inadequate nutrition education, they may prefer to buy the processed foods in a lower price range, which may contain high calories, fat, high sodium, or high sugar, instead of buying well-balanced meals [56]. For food-insecure people, the difficulty of maintaining their nutritional dietary needs may affect their body weight, which leads to being overweight or obese [57]. The findings of the present study support the previous finding that the numbers of food-insecure participants who live in urban areas and who have excess weight were greater than those in the food-secure group. Furthermore, people with food insecurity may encounter sleep disturbance, which is also associated with an increase in systolic blood pressure. The systolic blood pressure may be a result of jobs involving heavy work or of continuous vigorous physical activity [58,59,60]. A population-based study in a rural area of China found that older age, unemployment, lower-income, disability, and chronic disease comorbidities were significant factors associated with an increased risk of poor sleep quality for both men and women [61]. The significant interactions with race/ethnicity indicate that the relationship between sleep complaints and marital status, income, and employment differs between groups for men, and the relationship with education differs between groups for women [62]. Food insecurity is related to “poor” sleep quality, which may develop from anxiety, stress, or feelings of uncertainty about providing food and other necessities for themselves and their families [6,7].

Moreover, a combination of biological and psychosocial factors is involved in the mechanism of the relationship between food insecurity and poor mental health [9,63]. Meanwhile, the relationship between depression or having depressive symptoms and the experience of sleep disturbance is also known to be closely linked [15]. Food insecurity is related to depressive symptoms [29,64,65], and having depressive symptoms increases the odds of experiencing sleep disturbance in adults [9,14,66]. Another explanation is the compensatory mechanism of leptin, which reduces appetite and increases energy expenditure through the hypothalamic receptors [67,68]. Low leptin levels are associated with increased body mass index, lower quality of sleep, and a higher propensity toward having depressive symptoms [66]. The presence of sleep disturbance is associated with the effect of high levels of ghrelin and low levels of leptin. On the other hand, lower quality of sleep also causes greater neuronal activation in response to food stimuli, which results in increased motivation to seek food to achieve a high-energy intake, particularly by eating energy-dense foods that are high in fat and sugar [66,69]. Food-insecure adults may have diets that are deficient in nutrients, such as folic acid and tryptophan [70,71], which could influence mood and immune functions [72,73,74], which, in turn, may have an effect on their sleep.

The present study has some limitations. First, the usage of a cross-sectional study limited us from seeing the causal relationship between variables. However, our data were nationally representative for almost a majority of the adult Indonesian population. Second, although we controlled for covariates in the analysis, there remain several sociodemographic factors, such as individual income or house environmental factors that may contribute to sleep disturbance, which we could not include. Third, the sleep disturbance questions in the IFLS survey did not provide more potential sources of sleep disturbance. Thus, we were unable to specifically explain the type of sleep disturbance that the participants experienced (e.g., insomnia, sleep apnea, duration of sleep, and the latency of sleep). The use of self-reported data for sleep disturbance and physical activity variables is likely to suffer from a response bias and may affect the study results [75]. However, the PROMIS self-reported sleep disturbance has been used and validated in former research among adults [20,21,22]. A possible bidirectional relationship between depressive symptoms and sleep disturbance that we could not test in both directions may also be a limitation of the present work. For the study’s purpose, we only focused on the relationship between the exposure (i.e., depressive symptom) and sleep disturbance (as the outcome) because of the general assumption that depression treatment would also resolve the associated symptoms, such as sleep disturbance [76]. Lastly, since we used the food consumption score in the food security assessment, although the outcome investigated may not fully represent food insecurity, this method has been used widely in former studies [29,77,78]. Therefore, the interpretation of the study result must be taken cautiously.

## 5. Conclusions

In conclusion, the results of this study suggest that sleep disturbance may be affected by food insecurity in adults and may later lead to serious health outcomes. A potential solution that could overcome this problem is to encourage food-insecure people to participate in nutritional education programs that are conducted by health experts that also incorporate advice about the benefits of sleep quality.

## Figures and Tables

**Figure 1 nutrients-12-03411-f001:**
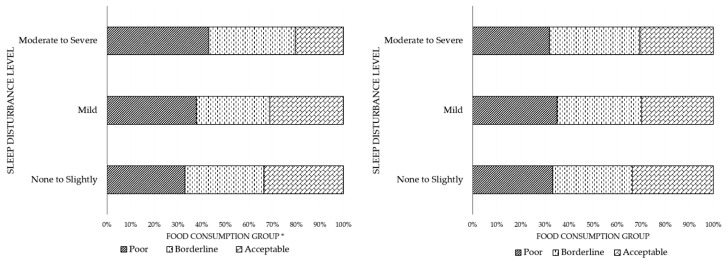
Prevalence of sleep disturbance among men (left) and women (right) by food consumption group. Indonesia Family Life Survey (IFLS) 2014 data were used in the analysis. * Significance: *p* < 0.05.

**Table 1 nutrients-12-03411-t001:** The participants’ characteristics.

Variable	All	Food-Secure	Food-Insecure	*p*-Value
*n*	20,212	9326 (46.14)	10,886 (53.86)	
Gender, *n* (%)				<0.001
Women	10,070 (49.82)	4468 (47.91)	5602 (51.46)	
Men	10,142 (50.18)	4858 (52.09)	5284 (48.54)	
Age (years), mean (SD)	39 (11)	39 (11)	38 (11)	0.015
Age Group (years), *n* (%)				0.037
≤35	9050 (44.78)	4088 (43.83)	4962 (45.58)	
36–55	9304 (46.03)	4378 (46.94)	4926 (45.25)	
≥56	1858 (9.19)	860 (9.22)	998 (9.17)	
Educational Attainment, *n* (%)				<0.001
Low (<12 years)	11,037 (54.61)	4205 (45.09)	6832 (62.76)	
High (≥12 years)	9175 (45.39)	5121 (54.91)	4054 (37.24)	
Marital Status, *n* (%)				0.001
Never Married	2328 (11.52)	1149 (12.32)	1179 (10.83)	
Currently or Ever Married	17,884 (88.48)	8177 (87.68)	9707 (89.17)	
Living Areas, *n* (%)				<0.001
Rural	8232 (40.73)	3387 (36.32)	4845 (44.51)	
Urban	11,980 (59.27)	5939 (63.68)	6041 (55.49)	
Smoking Habit, *n* (%)				<0.001
Never	12,148 (60.10)	5604 (60.09)	6544 (60.11)	
Current Smoker	879 (4.35)	481 (5.16)	398 (3.66)	
Former Smoker	7185 (35.55)	3241 (34.75)	3944 (36.23)	
Hypertension Medication User, *n* (%)				0.027
No	19,817 (98.05)	9122 (97.81)	10,695 (98.25)	
Yes	395 (1.95)	204 (2.19)	191 (1.75)	
Cholesterol Medication User, *n* (%)				<0.001
No	20,051 (99.20)	9226 (98.93)	10,825 (99.44)	
Yes	161 (0.80)	100 (1.07)	61 (0.56)	
Abdominal Obesity ^a^, *n* (%)				0.002
No	4226 (48.46)	1927 (46.69)	2299 (50.05)	
Yes	4494 (51.54)	2200 (53.31)	2294 (49.95)	
Body Mass Index (kg/m^2^), mean (SD)	24.28 (4.11)	24.39 (4.06)	24.19 (4.14)	0.001
Body Mass Index Classification ^b^, *n* (%)				0.013
18.5–25.0	12,664 (57.00)	5749 (61.64)	6915 (63.52)	
25.1–27.0	2923 (13.16)	1407 (15.09)	1516 (13.93)	
>27.0	4625 (20.82)	2170 (23.27)	2455 (22.55)	
Hypertension, *n* (%)				0.542
No	13,689 (67.73)	6296 (67.51)	7393 (67.91)	
Yes	6523 (32.27)	3030 (32.49)	3493 (32.09)	
Sleep Disturbance Level ^c^, *n* (%)				0.002
None to Slight	18,895 (93.48)	8772 (94.06)	10,123 (92.99)	
Mild	1050 (5.19)	455 (4.88)	595 (5.47)	
Moderate to Severe	267 (1.32)	99 (1.06)	168 (1.54)	
Depressive Symptoms ^d^, *n* (%)				<0.001
No	12,836 (63.51)	5758 (61.74)	7078 (65.02)	
Yes	7376 (36.49)	3568 (38.26)	3808 (34.98)	
Body Shape Index (m^11/6^ kg^−2/3^), mean (SD)	0.0810 (0.0057)	0.0810 (0.0055)	0.0810 (0.0059)	0.708
Waist Circumference (cm), mean (SD)	85.55 (10.95)	86.19 (10.84)	84.96 (11.03)	<0.001
Systolic Blood Pressures (mmHg), mean (SD)	129.56 (19.76)	129.18 (19.35)	129.89 (20.09)	0.010
Diastolic Blood Pressures (mmHg), mean (SD)	80.04 (11.97)	80.01 (12.01)	80.08 (11.94)	0.667
Food Consumption Score, mean (SD)	34.70 (14.66)	46.70 (10.45)	24.42 (8.75)	<0.001
Moderate PA Volume (METs min/w), mean (SD)	1145.88 (1872.15)	1185.25 (1907.31)	1112.16 (1840.92)	0.943
Vigorous PA Volume (METs min/w), mean (SD)	1190.33 (3136.19)	1093.61 (2950.50)	1273.19 (3284.81)	<0.001
Sleep Disturbance Score, mean (SD)	40.40 (11.51)	40.70 (11.08)	40.14 (11.86)	0.001
CES-D-10 Score, mean (SD)	8.70 (5.00)	9.00 (4.86)	8.45 (5.10)	<0.001

Abbreviations: SD, standard deviation; METs min/w, metabolic equivalent of tasks for minutes per week; CES-D-10, 10 items of the Center for Epidemiological Studies Depression questionnaire; PA, physical activity. Note: The categorical data are presented using *n* (%) and the continuous data are presented using mean (SD). ^a^ The definition of abdominal obesity for women and men used the waist circumference with cut-off points of >80 cm or >90 cm, respectively. ^b^ The body mass index used the adult categorization of body mass index for the Indonesian population. ^c^ The definition of having depressive symptoms used the score of the CES-D-10 with cut-off values ≥10. ^d^ The sleep disturbance level was defined using the t-score of the Patient-Reported Outcomes Measurement Information System (PROMIS) guidelines of the sleep disturbance questionnaire. Significance was set to *p* < 0.05.

**Table 2 nutrients-12-03411-t002:** Characteristics related to the sleep disturbance level, as determined by the regression model.

Variables	Sleep Disturbance *		
	OR	95% CI	*p*-Value
Gender (Ref: Men)			
Women	1.01	(0.68, 1.48)	0.972
Age (years, Ref: ≤35)			
36–55	1.35	(1.00, 1.82)	0.050
≥56	1.78	(1.17, 2.72)	0.007
Educational attainment (Ref: ≥12 years)			
Low (<12 years)	0.41	(0.30, 0.57)	<0.001
Marital Status (Ref: Never Married)			
Currently or Ever Married	1.49	(0.85, 2.62)	0.165
Living Areas (Ref: Rural)			
Urban	1.01	(0.78, 1.30)	0.949
Smoking Habit (Ref: Never)			
Current Smoker	0.48	(0.20, 1.15)	0.100
Former Smoker	0.81	(0.54, 1.21)	0.297
BMI (kg/m^2^, Ref: 18.5–25.0)			
25.1–27.0	0.95	(0.66, 1.37)	0.793
>27.0	0.83	(0.60, 1.15)	0.256
Depressive Symptoms (Ref: No)			
Yes	3.57	(2.77, 4.61)	<0.001
Food Security Status (Ref: Food-Secure)			
Food-Insecure	1.32	(1.02, 1.70)	0.036

Abbreviations: Ref., reference; OR, odds ratio; 95% CI, 95% confidence interval. Note: * Sleep disturbance was categorized into greater than mild or not greater than mild. The models were adjusted for age, gender, body mass index, education attainment, marital status, living area, smoking habits, physical activity volume, blood pressure value, food consumption score, and CES-D-10 score. Statistical significance was set to *p* < 0.05.

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
