# Peer review of "The Relationship between Food Security Status and Sleep Disturbance among Adults: A Cross-Sectional Study in an Indonesian Population"

_nutrients, 2020, doi:10.3390/nu12113411_

Round 1

Reviewer 1 Report

General comments: This is an interesting paper using data from a large population-based cross-sectional study of Indonesian adults conducted in 2014 which had adequate data on dietary intakes using a validated FFQ as well as measures of food security status and sleep. The methods used are sound. There are few comments that need to be addressed. 

Specific comments:

1) There are numerous grammatical and spelling errors throughout that need to be corrected. Copy-editing is definitely required. 

2) Bi-directional relationships should be labelled as such when the authors are trying to make the point that exposure--> outcome could also be outcome--> exposure. This should also be added to the study limitations in more detail. 

3) Limitations regarding self-reports vs. objective measures should be included. This can apply to physical activity and sleep. 

4) Some of the covariates included in the models can in fact be mediating the association between food insecurity and sleep. Thus, modeling should be done in steps, in which only exogenous variables are added to Model 1 (e.g age, sex, income) and then lifestyle and health-related factors are added in related clusters. This can be included as an Appendix. 

5) A flowchart showing how the final sample was selected is needed and some assessment of selection bias as well, particularly with respect to data missingness on exposures, outcomes and covariates. If those variables are missing at a rate <15% each, or about 15% on average, then multiple imputation can help determine whether there was any selection bias. Five imputations may be sufficient in this case. Otherwise, selection bias should be added as a potential limitation. 

Author Response

Response to the Reviewers

The Relationship between Food Security Status and Sleep Disturbance among Adults: A Cross-Sectional Study in an Indonesian Population

Dear all Reviewers,

We had revised our manuscript based on your helpful and constructive comments and suggestions as follow:

§  REVIEWER 1

Comment 1:

General comments: This is an interesting paper using data from a large population-based cross-sectional study of Indonesian adults conducted in 2014, which had adequate data on dietary intakes using a validated FFQ as well as measures of food security status and sleep. The methods used are sound. There are few comments that need to be addressed.

Specific comments:

1) There are numerous grammatical and spelling errors throughout that need to be corrected. Copy-editing is definitely required.

Response 1:

Thank you for your comments and constructive suggestions. We sent the manuscript for the English editing using the English editing services from the MDPI Company and followed by the certificate of proof. We had revised the manuscript accordingly. The current manuscript is the latest version after passed the English editing processes.

Comment 2:

2) Bi-directional relationships should be labelled as such when the authors are trying to make the point that exposure--> outcome could also be outcome--> exposure. This should also be added to the study limitations in more detail.

Response 2:

Thank you for the comments on the study limitations. We added the information related to the bidirectional relationship in the discussion part, which can be addressed on page 9 line 327-333.

“A possible bi-directional relationship between the depressive symptoms and sleep disturbance that we might not test in both directions may also become one limitation of the recent work. For the study purpose, we only focused on the relationship between the exposure (i.e., depressive symptom) and sleep disturbance (as the outcome) because of the general assumption that in depression treatment would also be resolved the associated symptom such as sleep disturbance [1]. Therefore, the interpretation of the study result must be taken cautiously.”

Comment 3:

3) Limitations regarding self-reports vs. objective measures should be included. This can apply to physical activity and sleep.

Response 3:

We revised the discussion part and added the information regarding the limitation of the study and some explanation about the self-reported variables. These can be addressed on the manuscript on page 9 line 325-327.

“The use of self-reported data for sleep disturbance and physical activity variables is likely to be a response bias and may affect the study results [2]. However, the PROMIS self-reported sleep disturbance has been used and validated in former research among adults [3-5].”

Comment 4:

4) Some of the covariates included in the models can in fact be mediating the association between food insecurity and sleep. Thus, modeling should be done in steps, in which only exogenous variables are added to Model 1 (e.g., age, sex, income) and then lifestyle and health-related factors are added in related clusters. This can be included as an Appendix.

Response 4:

Thank you for your suggestions. We had addressed the results section about the covariates included in the model. We added these parts in the supplementary materials of the manuscripts as follow:

Table S 1. The logistic regression between the covariate variables and the outcome.

Variables

Sleep Disturbance * M1

OR

95% CI

p value

Gender (ref: men)

Women

1.17

(0.92, 1.49)

0.201

Age (years, ref: ≤ 35)

36 - 55

1.03

(0.63, 1.67)

0.909

≥ 56

1.16

(0.50, 2.69)

0.728

Educational attainment (ref: ≥ 12 years)

Low (< 12 years)

0.38

(0.28, 0.51)

< 0.001

Marital Status (ref: never married)

Currently or Ever Married

1.43

(0.82, 2.48)

0.206

Living Areas (ref: rural)

Urban

0.81

(0.64, 1.04)

0.096

Smoking Habit (ref: never)

Current Smoker

0.49

(0.20, 1.16)

0.104

Quit smoker

1.06

(0.72, 1.57)

0.763

BMI (kg/m2, ref: 18.5 - 25.0)

25.1 - 27.0

0.90

(0.63, 1.29)

0.571

> 27.0

0.79

(0.58, 1.08)

0.135

Depressive Symptoms (ref: no)

Yes

3.50

(2.72, 4.52)

< 0.001

Food Security Status (ref: food secure)

Food Insecure

1.47

(1.15, 1.89)

0.002

Abbreviations: OR, Odd Ratio; CI, 95% confidence interval; M1, Model 1. Note: * Sleep disturbance was defined as greater than mild or without greater than mild. Models are adjusted for age, gender, body mass index, education attainment, marital status, living areas, smoking habits, physical activity volumes, blood pressure values, and food consumption score, CES-D-10 Score. Model 1 is adjusted for age and gender. All statistically significant values are set to < 0.05.

We added some explanation (on page 3 line 241-246) related to the table S1 as follow, “Further, table S1 shows the higher likelihood of experiencing sleep disturbance after adjustment for age and gender in our study was reported in participants with depressive symptoms (OR = 3.50; 95% CI: 2.72-4.52; p < 0.001) and participants with food insecurity (OR = 1.47; 95% CI: 1.15-1.89; p = 0.002). On the other hand, the lower likelihood of experiencing sleep disturbance was reported in participants with low educational attainment (OR = 0.38; 95% CI: 0.28-0.51; p < 0.001).”

Comment 5:

5) A flowchart showing how the final sample was selected is needed and some assessment of selection bias as well, particularly with respect to data missingness on exposures, outcomes and covariates. If those variables are missing at a rate <15% each, or about 15% on average, then multiple imputation can help determine whether there was any selection bias. Five imputations may be sufficient in this case. Otherwise, selection bias should be added as a potential limitation.

Response 5:

We had addressed the results section about the flowchart of the selection sample of the population. We added this information as a supplementary file as well as on the result section page 4 line 180-181. The missing data on the sampling process was higher than the 15%, so we consider not using the multiple imputation in the process, instead deleted the one with missing data.

Figure S 1. Flowchart of the Sampling Participants

Reviewer 2 Report

  • This exposure assessment does not represent food insecurity, which can be defined as “the state of being without reliable access to a sufficient quantity of affordable, nutritious food.” Below is a sample question of one dimension or domain:
    • In the last 30 days, were you ever hungry but didn't eat because there wasn't enough money for food?
    • Food insecurity is not about “the person’s **failure to fulfill their need** for a nutritious diet. This seems like victim-blaming.
    • The manuscript’s framing may need to be modified based on the actual exposure measurement.
  • Relevant to the exposure not representing food insecurity, be specific and provide the details (of variable assessments) so the reader can properly assess the methodology.
  • Either Poisson regression with robust variance or negative binomial regression models should be used instead of logistic regression model was used. Otherwise, the results will be overestimated, but appears to be the case.
    • Justification: Barros AJ, Hirakata VN. Alternatives for logistic regression in cross-sectional studies: an empirical comparison of models that directly estimate the prevalence ratio. BMC Med Res Methodol. 2003;3:21.
  • Informed consent and IRB approval were not mentioned in this manuscript.
  • The authors need to list self-reported sleep as a limitation, and discuss how the self-reported nature of the data may have impacted their results. For instance, do you think the results were over- or under-estimated? Were validation studies conducted in this (or a similar) population?
  • Did the results differ by sex? By BMI? Sensitivity analyses can be conducted.
  • This is a mere preference, but I don't think third person should be used. The authors are the ones who conducted the study, right?

Author Response

Response to the Reviewers

The Relationship between Food Security Status and Sleep Disturbance among Adults: A Cross-Sectional Study in an Indonesian Population

Dear all Reviewers,

We had revised our manuscript based on your helpful and constructive comments and suggestions as follow:

§   REVIEWER 2

Comment 1:

This exposure assessment does not represent food insecurity, which can be defined as “the state of being without reliable access to a sufficient quantity of affordable, nutritious food.” Below is a sample question of one dimension or domain: “In the last 30 days, were you ever hungry but did not eat because there was not enough money for food?”

Food insecurity is not about “the person’s **failure to fulfill their need** for a nutritious diet. This seems like victim-blaming.

The manuscript’s framing may need to be modified based on the actual exposure measurement. Relevant to the exposure not representing food insecurity, be specific and provide the details (of variable assessments) so the reader can properly assess the methodology.

Response 1:

Thank you for the comments. We revised the definition of the food insecurity into:

“The assessment of food insecurity is associated with a person’s lack of secure access to fulfill their need for a nutritious diet in a sufficient amount to keep an active healthy life, considering both the food frequency and food diversity.”

This part can be addressed on the main text of the manuscript on page 3 line 96-98. The indicators of food insecurity can be approached from several ways, such as food access/availability or dietary diversity [6]. Yet, the hunger concept is associated with food insecurity, which referred to as food deprivation, but it is not the only domain for assessing food insecurity. There also some conditions of the food insecure people are not hungry because there are other causes including the poor choice of food that may lead to the poor intake of micronutrients. Thus, for the purpose of our study, we approached the food insecurity assessment from the dietary diversity and the frequency of the participants that can be assessed using the food frequency questionnaire. The former researchers have used the assessment of the food insecurity status using the WFP of food insecurity concept widely [7,8].

Comment 2:

Either Poisson regression with robust variance or negative binomial regression models should be used instead of logistic regression model was used. Otherwise, the results will be overestimated, but appears to be the case.

Justification: Barros AJ, Hirakata VN. Alternatives for logistic regression in cross-sectional studies: an empirical comparison of models that directly estimate the prevalence ratio. BMC Med ResMethodol. 2003; 3:21.

Response 2:

Thank you for your constructive comments and suggestion. We had done the Poisson regression as suggested earlier. The table of the Poisson regression using the robust variance is presented as follow:

Table R 1. The Logistic, Poisson, and Negative Binomial Regression between the Covariates and the Sleep Disturbance

Variables

Sleep Disturbance *

Sleep Disturbance * P

Sleep Disturbance * NB

OR

95% CI

p value

IRR

95% CI

p value

IRR

95% CI

p value

Gender (ref: men)

Women

1.01

(0.68, 1.48)

0.972

1.01

(0.69, 1.47)

0.976

1.01

(0.69, 1.47)

0.976

Age (years, ref: ≤ 35)

36 - 55

1.35

(1.00, 1.82)

0.050

1.33

(1.01, 1.77)

0.044

1.33

(0.99, 1.79)

0.054

≥ 56

1.78

(1.17, 2.72)

0.007

1.74

(1.16, 2.59)

0.007

1.74

(1.15, 2.62)

0.009

Educational attainment (ref: ≥ 12 years)

Low (< 12 years)

0.41

(0.30, 0.57)

< 0.001

0.42

(0.31, 0.57)

< 0.001

0.42

(0.31, 0.58)

< 0.001

Marital Status (ref: never married)

Currently or Ever Married

1.49

(0.85, 2.62)

0.165

1.48

(0.85, 2.55)

0.164

1.48

(0.84, 2.58)

0.173

Living Areas (ref: rural)

Urban

1.01

(0.78, 1.30)

0.949

1.01

(0.79, 1.29)

0.931

1.01

(0.79, 1.30)

0.933

Smoking Habit (ref: never)

Current Smoker

0.48

(0.20, 1.15)

0.100

0.49

(0.20, 1.21)

0.121

0.49

(0.21, 1.16)

0.106

Quit smoker

0.81

(0.54, 1.21)

0.297

0.82

(0.55, 1.21)

0.317

0.82

(0.55, 1.20)

0.309

BMI (kg/m2, ref: 18.5 - 25.0)

25.1 - 27.0

0.95

(0.66, 1.37)

0.793

0.95

(0.67, 1.35)

0.793

0.95

(0.67, 1.36)

0.795

> 27.0

0.83

(0.60, 1.15)

0.256

0.83

(0.61, 1.14)

0.263

0.83

(0.61, 1.15)

0.271

Depressive Symptoms (ref: no)

Yes

3.57

(2.77, 4.61)

< 0.001

3.47

(2.72, 4.43)

< 0.001

3.47

(2.70, 4.47)

< 0.001

Food Security Status (ref: food secure)

Food Insecure

1.32

(1.02, 1.70)

0.036

1.30

(1.02, 1.67)

0.037

1.30

(1.01, 1.67)

0.041

Abbreviations: OR, Odd Ratio; CI, 95% confidence interval; P, Poisson Regression, NB, Negative binomial Regression. Note: * Sleep disturbance was defined as greater than mild or without greater than mild. Models are adjusted for age, gender, body mass index, education attainment, marital status, living areas, smoking habits, physical activity volumes, blood pressure values, and food consumption score, CES-D-10 Score. The Poisson Regression was used the robust variance and the results show as the incidence rate-ratio. All statistically significant values are set to < 0.05.

Table R1 shows the logistic, Poisson, and negative binomial regression between the covariates and the sleep disturbance. The logistic regression is presented as odd ratio values while the Poisson and negative binomial regression are presented as the incidence rate ratios. The Poisson and negative binomial regression models are widely used in dealing the number of occurrences (counts) of an event. The basic assumption of Poisson regression are, first, there is a quantity called the incidence rate that is the rate at which events occur, second, the incidence rate can be multiplied by exposure to obtain the expected number of observed events. Third, over very small exposures, the probability of finding more than one event is small compared with, and forth, non-overlapping exposures are mutually independent [9]. The limitation of Poisson regression modeling, such as the assumption of the conditional distribution of the outcome variable that require mean and variance to be equal. In the analysis of the contagious events or as a results when the success outcome is not rare, the overdispersion is expected to be happened. The negative binomial regression is commonly used when the user have to deal with the limitation of the Poisson regression [10,11].

Based on the table R1 and some introduction of the usage of the Poisson and negative binomial regression models, we consider using the logistic regression to test our hypotheses. The outcome in our current study was a binary data which contains the two conditions of the participants, whether they have a greater than mild (labelled as “0”) or without greater than mild (labelled as “1”) experience of sleep distribution. The self-reported sleep disturbance variable was assessed one time when the participants filling the questionnaire and did not have a record for some individual repeated events. Thus, our data was not suit the requirement to use the Poisson and negative binomial regression models.

Comment 3:

Informed consent and IRB approval were not mentioned in this manuscript.

Response 3:

Thank you for the comments. We added the description related to the informed consent and IRB approval on the material and method section of the manuscript that can be addressed on page 2 line 71-74.

 “The institutional review board (IRB) review of the IFLS studies went through the sufficient and appropriate review that followed the IRB guidelines and was approved by both the RAND Corporation and Indonesia's Institutions, in particular Survey Meter institution for the IFLS5 study [12,13].”

Comment 4:

The authors need to list self-reported sleep as a limitation, and discuss how the self-reported nature of the data may have impacted their results. For instance, do you think the results were over- or under-estimated? Were validation studies conducted in this (or a similar) population?

Response 4:

Thank you for your suggestion. We revised the discussion part and added the information regarding the limitation of the study and some explanation about the self-reported of sleep disturbance variable. These can be addressed on the manuscript on page 9 line 325-327.

“The use of self-reported data for sleep disturbance and physical activity variables is likely to be a response bias and may affect the interpretation of the study results [2]. However, the PROMIS self-reported sleep disturbance has been used and validated in former research among adults [3-5].”

Comment 5:

Did the results differ by sex? By BMI? Sensitivity analyses can be conducted.

Response 5:

We conducted the analyses to support the results that stratified by gender and we added it on the supplementary materials as table S2. The logistic regression between characteristics related to the sleep disturbance level.

Variables

Sleep Disturbance (Men)

Sleep Disturbance (Women)

OR

95% CI

p value

OR

95% CI

p value

Age (years, ref: ≤ 35)

36 - 55

1.04

(0.68, 1.57)

0.866

1.73

(1.00, 1.82)

0.016

≥ 56

1.26

(0.65, 2.40)

0.493

2.54

(1.17, 2.72)

0.002

Educational attainment (ref: ≥ 12 years)

Low (< 12 years)

0.46

(0.30, 0.71)

< 0.001

0.37

(0.22, 0.61)

< 0.001

Marital Status (ref: never married)

Currently or Ever Married

1.64

(0.82, 3.25)

0.158

1.61

(0.57, 4.55)

0.374

Living Areas (ref: rural)

Urban

1.13

(0.78, 1.65)

0.526

0.88

(0.62, 1.25)

0.483

Smoking Habit (ref: never)

Current Smoker

0.35

(0.12, 0.99)

0.048

2.12

(0.47, 9.55)

0.329

Quit smoker

0.78

(0.50, 1.20)

0.251

0.78

(0.30, 2.02)

0.605

BMI (kg/m2, ref: 18.5 - 25.0)

25.1 - 27.0

1.50

(0.90, 2.49)

0.121

0.66

(0.40, 1.11)

0.118

> 27.0

0.90

(0.50, 1.63)

0.732

0.77

(0.52, 1.15)

0.201

Depressive Symptoms (ref: no)

Yes

3.16

(2.17, 4.60)

< 0.001

3.93

(2.77, 5.57)

< 0.001

Food Security Status (ref: food secure)

Food Insecure

1.75

(1.19, 2.58)

0.005

1.07

(0.76, 1.51)

0.706

Abbreviations: OR, Odd Ratio; CI, 95% confidence interval. Note: * Sleep disturbance was defined as greater than mild or without greater than mild. Models are adjusted for age, gender, body mass index, education attainment, marital status, living areas, smoking habits, physical activity volumes, blood pressure values, and food consumption score, CES-D-10 Score.All statistically significant values are set to < 0.05.

We also added some explanation on the results section (page 7-8 line 246-257) related to the table S2.

“The higher likelihood of experiencing sleep disturbance among men in our study was reported in participants with depressive symptoms (OR = 3.16; 95% CI: 2.17-4.60; p < 0.001) and participants with food insecurity (OR = 1.75; 95% CI: 1.19-2.58; p = 0.005). Meanwhile, the higher likelihood of experiencing sleep disturbance among women in our study was reported in participants with depressive symptoms (OR = 3.16; 95% CI: 2.17-4.60; p < 0.001) and in participants aged older than 36 years (OR = 1.73-2.54; 95% CI: 1.00-2.72; p = 0.016-0.002). On the other hand, the lower likelihood of experiencing sleep disturbance among men in our study was reported in current smoker participants (OR = 0.35; 95% CI: 0.12-0.99; p = 0.048) and in participants with low educational attainment (OR = 0.46; 95% CI: 0.30-0.71; p < 0.001). Moreover, the lower likelihood of experiencing sleep disturbance among women in our study was reported in participants with low educational attainment (OR = 0.37; 95% CI: 0.22-0.61; p < 0.001).”

Comment 6:

This is a mere preference, but I don't think third person should be used. The authors are the ones who conducted the study, right?

Response 6:

Thank you for the comments on this subject. We had revised the words that use the third person communication style on our manuscript from “the authors” into “we”.

Round 2

Reviewer 1 Report

The manuscript is greatly improved. I have no further comments. 
